# Methamphetamine Increases the Proportion of SIV-Infected Microglia/Macrophages, Alters Metabolic Pathways, and Elevates Cell Death Pathways: A Single-Cell Analysis

**DOI:** 10.3390/v12111297

**Published:** 2020-11-12

**Authors:** Meng Niu, Brenda Morsey, Benjamin G. Lamberty, Katy Emanuel, Fang Yu, Rosiris León-Rivera, Joan W. Berman, Peter J. Gaskill, Stephanie M. Matt, Pawel S. Ciborowski, Howard S. Fox

**Affiliations:** 1Department of Genetics, Cell Biology and Anatomy, University of Nebraska Medical Center, Omaha, NE 68198, USA; meng.niu@unmc.edu; 2Department of Neurological Sciences, University of Nebraska Medical Center, Omaha, NE 68198, USA; bmorsey@unmc.edu (B.M.); bglamberty@unmc.edu (B.G.L.); katy.emanuel@unmc.edu (K.E.); 3Department of Biostatistics, University of Nebraska Medical Center, Omaha, NE 68198, USA; fangyu@unmc.edu; 4Department of Pathology, Albert Einstein College of Medicine, Bronx, NY 10461, USA; rosiris.leonrivera@einsteinmed.org (R.L.-R.); joan.berman@einsteinmed.org (J.W.B.); 5Department of Pharmacology and Physiology, Drexel University College of Medicine, Philadelphia, PA 19129, USA; pjg63@drexel.edu (P.J.G.); smm678@drexel.edu (S.M.M.); 6Department of Pharmacology and Experimental Neuroscience, University of Nebraska Medical Center, Omaha, NE 68198, USA; pciborowski@unmc.edu

**Keywords:** HIV, SIV, neuroHIV, HIV-associated neurocognitive disorders (HAND), macrophage, microglia, immunometabolism, scRNA-seq, BDNF

## Abstract

Both substance use disorder and HIV infection continue to affect many individuals. Both have untoward effects on the brain, and the two conditions often co-exist. In the brain, macrophages and microglia are infectable by HIV, and these cells are also targets for the effects of drugs of abuse, such as the psychostimulant methamphetamine. To determine the interaction of HIV and methamphetamine, we isolated microglia and brain macrophages from SIV-infected rhesus monkeys that were treated with or without methamphetamine. Cells were subjected to single-cell RNA sequencing and results were analyzed by statistical and bioinformatic analysis. In the animals treated with methamphetamine, a significantly increased proportion of the microglia and/or macrophages were infected by SIV. In addition, gene encoding functions in cell death pathways were increased, and the brain-derived neurotropic factor pathway was inhibited. The gene expression patterns in infected cells did not cluster separately from uninfected cells, but clusters comprised of microglia and/or macrophages from methamphetamine-treated animals differed in neuroinflammatory and metabolic pathways from those comprised of cells from untreated animals. Methamphetamine increases CNS infection by SIV and has adverse effects on both infected and uninfected microglia and brain macrophages, highlighting the dual and interacting harms of HIV infection and drug abuse on the brain.

## 1. Introduction

Methamphetamine (meth) is a highly addictive psychostimulant and neurotoxicant [1,2], and its use can lead to neuroinflammatory responses in the brain [3,4,5,6,7]. Meth use is associated with risky behaviors, helping to explain the high prevalence of human immunodeficiency virus (HIV) in meth users [8,9,10,11,12,13,14,15,16,17,18]. While HIV is primarily known for its impairment of the immune system, it also infects and affects the central nervous system (CNS). Within the CNS, microglia and macrophages are the main cells productively infected by HIV, as well as by simian immunodeficiency virus, SIV, which results in simian AIDS in susceptible nonhuman primates. These long-lived cells can act as viral reservoirs and serve as the source of chronic infection [19]. The impact of CNS infection is substantial, as studies in the US and Europe have revealed continued persistence of HIV-associated neurocognitive disorders (HAND), also known as neuroHIV, affecting up to 50% of people living with HIV (PLWH) [20,21,22]. Unfortunately, antiretroviral treatment does not completely shield the brain from the effects of HIV infection. Furthermore, many of the widely used drugs do not penetrate effectively into the CNS [23,24,25].

A number of untoward effects of meth has been found in PLWH [26,27,28,29], and meth can substantively worsen clinical outcomes in PLWH and accelerate systemic disease [30,31,32]. This includes the brain, where meth worsens neurocognitive outcomes, neuropathology, and neuropsychiatric disease [33,34,35,36,37]. In brains of both humans and the nonhuman primate model system, as well as in vitro, meth can affect HIV/SIV infection of both macrophages and microglia [38,39,40]. The need to understand these effects is important, as the use of meth remains a significant problem in the HIV-infected population. There has been a continued increase in deaths due to meth overdose since 2014 [41], and there has been an almost 500% increase in drug test positivity rates for methamphetamine from 2013 to 2019 [42]. The meth and opioid use epidemics have begun to overlap [43,44], which could further increase transmission of HIV. These data underscore the persistent and growing problems associated with meth abuse, and the need to understand its effects as it relates to neuroHIV.

The development of single-cell RNA sequencing (scRNA-seq) has led to striking advancements, enabling cell types to be characterized using the entire transcriptome of thousands of individual cells, providing a higher resolution of cellular differences and a better understanding of the normal and pathological functions of individual cells in numerous physiological systems. Recently, scRNA-seq has demonstrated facets of the development and diversity of microglia and brain macrophages as well as their responsiveness in vivo in disease models and neuroinflammatory stimulation [45,46,47,48,49,50,51,52,53]. Given the intersection of meth and HIV in PLWH, and the important role of microglia and brain macrophages in HIV/SIV neuropathogenesis, we examined the effect of a chronic meth administration regimen on microglia and brain macrophages in SIV-infected rhesus monkeys that developed neurological disease. Meth had pronounced effects, increasing the proportion of SIV-infected cells and altering their gene expression phenotype and predicted functional properties.

## 2. Materials and Methods

### 2.1. Animal Samples, SIV Infection, and Methamphetamine Treatment

All samples were derived from cryopreserved specimens from prior animal work, which was performed in an AAALAC-approved facility with UNMC Institutional Animal Care and Use Committee approval (protocol 11-032-05FC, first approved 20-May-2011, most recent approval 19-Dec-2018). All animal work followed National Institutes of Health and US Department of Agriculture guidelines. For the scRNA-seq from infected animals, specimens were from four rhesus monkeys that were inoculated intravenously with a virus stock derived from SIV_mac_251. For the plasma analysis, specimens were from 13 uninfected animals.

For meth treatment, animals were acclimated to the drug using an escalating protocol, where the concentration of meth was slowly increased over a month-long period (0.1–2.5 mg/kg of meth dissolved in sterile saline, delivered intramuscularly; Sigma Aldrich, St. Louis, MO, USA). After the ramp-up phase, animals were maintained at 2.5 mg/kg using a five-day on and two-day off paradigm for the entire study period. Untreated animals were injected with saline following the same injection regime. For the infected animals, treated animals were maintained for 60 days on 2.5 mg/kg of meth before SIV infection, and treatment continued after infection until the day of necropsy. For the uninfected animals, seven animals were treated with meth and six received saline as a control; blood samples were taken after 56 days of maintenance for assessment.

Necropsies were performed as determined by the animal protocol, which necessitated sacrifice due to disease progression and exhibiting symptoms suggestive of viral encephalitis. At necropsy, deeply anesthetized animals were perfused intracardially with sterile PBS containing 1 U/mL of heparin to clear blood including blood-borne cells from the brains, and brains harvested for histopathology, viral determination, and further experimental studies. A cellular preparation enriched in microglia and inflammatory cells was isolated through our published Percoll gradient technique [54]; these microglia/macrophage-enriched isolates were cryopreserved in 10% dimethyl sulfoxide in fetal bovine serum (FBS, Sigma Aldrich) and kept frozen at −140 °C.

For viral load determinations, EDTA-anticoagulated plasma was separated from blood by centrifugation. Tissue RNA was isolated by the Trizol method (Invitrogen, Carlsbad, CA, USA). SIV RNA was measured using the branched DNA assay by Siemens (Emeryville, CA, USA). For brain-derived neurotropic factor (BDNF) determination, EDTA-anticoagulated plasma was assayed using an immunoassay by Myriad RBM (Austin, TX, USA).

### 2.2. In Situ Hybridization

In situ hybridization for SIV RNA was performed using RNAscope (Advanced Cell Diagnostics, Newark, CA, USA) with probes targeting SIV (Cat No. 317221, which targets SIV coding sequences of gag, pol, vif, vpx, vpr, nef, and tat) and the 2.5 HD RED Assay. The manufacturer’s protocol was followed, with the exception that the probes were diluted 1:5 in the probe diluent and amplification 5 and 6 were decreased by 10 min and 5 min, respectively.

### 2.3. Single Cell Preparation and RNA Sequencing

Samples of cryopreserved enriched brain microglia/macrophages, described above, were rapidly thawed and transferred to RPMI with 10% FBS, supplemented with 1% DNAse (Sigma Aldrich), and allowed to rest at room temperature for 10 min. Cells were transferred to warm RPMI with 10% FBS and incubated at 37 °C for 15 min. Cells were washed and counted for use. Cells were stained for live vs. dead assessment with UV blue live dead assay (Invitrogen), and single cells were purified using immunomagnetic isolation with CD11b-conjugated non-human primate microbeads (Miltenyi, San Diego, CA, USA), followed by staining with CD11b BV605 clone M170 (Biolegend, San Diego, CA, USA). Cells were sorted based on size, singlets, live, and CD11b+ events using an Aria2 flow cytometer (BD Biosciences, San Jose, CA, USA).

Post-sorting, isolates were concentrated to approximately 1000 cells per µL, assessed by trypan blue for viability and concentration. Based on 10× Genomics parameters targeting 8000 cells, the ideal volume of cells was loaded onto the 10× Genomics (Pleasanton, CA, USA) Chromium GEM Chip and placed into the Chromium Controller for cell capturing and library preparation. This occurs through microfluidics and combining with Single Cell 3′ Gel Beads containing unique barcoded primers with a unique molecular identifier (UMI), followed by lysis of cells and barcoded reverse transcription of RNA, amplification of barcoded cDNA, fragmentation of cDNA to 200 bp 5′ adapter attachment, and sample indexing as the manufacturer instructed with version 3 reagent kits. The prepared libraries were then sequenced using an Illumina (San Diego, CA, USA) Nextseq550 sequencer.

### 2.4. Bioinformatic Analyses

The 10x Genomics Cellranger version 3.1 analysis pipeline was used for demultiplexing and generating the filtered feature barcode matrices. Briefly, cellranger mkfastq was used to convert the raw base call (BCL) files of scRNA-seq into FASTQ files. Then cellranger count was used to aligned reads to the custom combined genomes of *Macaca mulatta* (Mmul 10) and a novel five-section SIV genome (using NBCI reference sequence M33262.1). Similar to our recently published report [55], we divided the SIV genome into five sections termed SIV_A through SIV_E, creating gene annotations for well-characterized SIV splice sites (see Figure 1). Cellranger count also performed filtering, barcode counting, and UMI counting. As a result, a gene expression matrix was generated containing the raw UMI counts for each cell for each sample. The metrics for the sequencing and alignment are presented in Appendix A.

Partek (St. Louis, MO, USA) Flow version 9.0 was used for in-depth analysis, with the gene expression matrices for each sample as the input. The single cell QA/QC was performed to filter out low-quality cells based on the Total UMI count (<700 or >14,000), Detected Gene count (<400 or >4000), and Mitochondrial UMI proportion (>10%), resulting in the removal of 1692 cells, leaving 25,131. This was followed by a noise reduction step, in which the genes that had zero expression in all cells were removed from further analysis. To ensure an accurate annotation, we next created a strict mapping list from monkey to human genes. Biomart [56,57] was used to extract the list of gene to gene mapping from monkey to human, with confidence scores equal to 1 and alignment coverage over 75%. Non-protein coding genes, such as ribosomal RNA, miRNA, snRNA, lncRNA, and other genes, were removed, in addition to genes that coded for ribosomal proteins and mitochondrial-encoded proteins. The resulting list was then filtered by the monkey gene IDs from the Mmul10 reference genome. A final list of 15,770 unique monkey to human gene mappings was created, and the list of monkey genes in this mapping, plus the five SIV gene segments, was used as a feature filter in our subsequent work.

Normalization was performed by converting the UMI to CPM (counts per million reads), then adding 1 and converting to the log_2_ value. The data were then filtered by the combination of annotated rhesus gene and SIV gene list. Principle component analysis (PCA) was performed with a setting of 50 PCs for calculation. Graph-based clustering was performed with a resolution of 0.1 to achieve maximal modularity, nearest neighbors of 30, a scale of 100,000, the first 18 PCs from the PCA (according to the PC scree plot), and Louvain as the clustering algorithm. Data visualization was performed using Uniform Manifold Approximation and Projection (UMAP), with a local neighborhood size of 15, a minimal distance of 0.1, Euclidean as the distance metric, and the first 18 PCs from the PCA. Differentially expressed genes (DEGs) analysis was performed using Gene Specific Analysis (GSA). Hierarchical clustering was performed on the DEGs identified between SIVE and Meth-SIVE, which were filtered for total counts > 500,000 to eliminate genes with low expression levels. Average Linkage was used as the cluster distance metric and Euclidean as the point distance metric.

Additional bioinformatic analysis, using the human gene IDs from the conversion described above, was performed using Ingenuity Pathway Analysis (IPA) (QIAGEN, Germantown, MD, USA). We provided the DEG lists (filtered for total counts > 500,000 as above) to IPA for its core analysis and then overlaid with the global molecular network in the Ingenuity pathway knowledge base. As a result, the biological functions, disease annotations, and canonical pathways that were enriched in the DEG lists were identified and analyzed.

### 2.5. Statistics

Alpha was set to <0.05 for all analyses reported. Gene expression values are expressed as the least square mean of the population. DEGs were defined as those with a fold change of >|1.5| and a false discovery rate (FDR) < 0.05. FDRs for DEGs were calculated using Partek, and *p*-values for biologic functions and disease, as well as canonical pathways by IPA, are fully reported in the Appendix A. Z-scores were calculated by IPA. All bioinformatics analyses on differentially expressed genes from the conditions or graph-based clusters were performed on gene lists filtered to eliminate lowly expressed genes (with total counts < 500,000), and as above selected for those with a fold change of >|1.5| and FDR < 0.05. Chi-square tests were used to compare the positive rates between the two conditions for infected cells and for each viral genomic region, as well as to compare the positive rates among the five clusters for infected cells and on each region. Pairwise group comparisons between clusters were made if the overall test was significant, adjusting for multiple comparisons with Bonferroni’s method. The Mann–Whitney U test was used to compare levels of plasma BDNF.

### 2.6. Data Availability

The scRNA-seq data from this study has been deposited in the NCBI GEO database, accession # GSE160384.

## 3. Results

### 3.1. Characterization of Samples

Cells were obtained from cryopreserved samples of enriched microglia/macrophages from our prior studies. SIV encephalitis (SIVE), used as a model for the most severe form of HIV neuropathology, HIV encephalitis (HIVE), occurs more frequently in animals that develop a rapid disease course to simian AIDS after SIV infection [58]. Therefore, we chose cells from animals with such a course leading to SIVE, two of whom were treated with meth before and during infection, as described in Methods, and two who were not. Viral loads, measured in plasma, spleen, and brain regions, did not appreciably differ (Table 1), and in situ hybridization for SIV RNA revealed abundant expression in cells with the morphological appearance of macrophages and microglia (Figure 1). The cryopreserved cells were thawed, purified for CD11b positivity (expressed by monocytes, macrophages, and microglia, [59]) by sequential immunomagnetic and flow cytometric techniques, and processed for scRNA-seq using the 10× Genomics platform.

### 3.2. Single-Cell Cluster Analysis and SIV Infection

After quality assurance and quality control (QA/QC, filtering out 1692 cells), noise reduction (removing non-expressed genes), a normalization step, and filtering by the annotated gene list, we performed principal component analysis (PCA) on the 25,131 cells with the 13,734 annotated expressed genes. With the first 18 PCs, we performed graph-based clustering and projected the results on the UMAP as shown in Figure 2, revealing seven clusters. Based on the top expressed gene in each of the clusters, we identified cluster 6 as likely containing cytotoxic T lymphocytes (CTLs) and natural killer (NK) cells from the enrichment of expression of marker genes (*GZMA*, *GZMB*, *NKG7*, *CD3D*, and *CD3G*), and cluster 7 as containing likely endothelial cells from enriched marker gene expression (*RGS5*, *CLDN5*, and *ATP1A2*) (Appendix A). The other five clusters expressed genes consistent with macrophages and microglia (see below).

Clusters 6 and 7 (comprising 927 cells) were then removed from the analysis to enable the evaluation of the macrophage/microglia, and PCA was performed on the remaining 24,204 cells. Again using the first 18 PCs, we performed graph-based clustering and projected the results on the UMAP as shown in Figure 3A. Five clusters were found. Interestingly, these clusters also reflected the treatment conditions (Figure 3B). Cluster 1 is largely cells from the SIVE samples, clusters 3 and 4 are largely cells from the Meth-SIVE samples, and clusters 2 and 5 are shared with SIVE and Meth-SIVE cells.

SIV transcripts were found in a total of 2419 cells (10% of the macrophage/microglia). Examination of these cells within the UMAP revealed that cells with SIV transcripts were distributed across both conditions and all clusters (Figure 3C). The lack of clustering of the SIV-positive cells separate from SIV-negative ones (exposed to the inflammatory milieu but without detectable SIV transcripts) was similar to what we recently reported in HIV-infected mature monocytes [55]. We then examined the distribution of the SIV transcripts by dividing the genome into five regions based on splice patterns (Figure 4), as well as examining all SIV-positive cells as a whole, in the two conditions and five clusters (Table 2). Chi-squared analysis indicated that the Meth-SIVE condition has a significantly higher proportion of SIV-positive cells than the SIVE condition, as did cluster 4 compared to the other clusters (followed by clusters 2 and 5, which were not statistically different from each other). Region B, which is only found in unspliced viral RNA, was highest in clusters 2, 4, and 5, whereas region D, which can be found in unspliced as well as singly spliced viral RNA, was highest in clusters 2 and 5. Complete analyses of the viral regions and cellular clusters can be found in Appendix A. The relative predominance of the SIV_E region is likely due to its presence in all transcripts, as well as its location near the polyadenylation site, as the reverse transcription for cDNA production was initiated by oligo-dT containing primers.

The regions of the SIV genome (SIV_A–E) are defined as shown in Figure 3. The number of nucleotides (bases) within those regions is indicated. For cells in the two conditions and five clusters, the percentages of cells that express transcripts within those regions, as well as a transcript in one or more of these regions (SIV_all), are given.

### 3.3. Effect of Meth on SIVE

#### 3.3.1. Analysis of Macrophage/Microglia Marker Genes

We then compared gene expression levels between the Meth-SIVE and SIVE conditions using gene specific analysis (GSA). We first assessed known macrophage/microglia marker genes in the two conditions and five clusters. Cells from the two conditions showed differential expression of a number of these genes (Figure 5, Appendix A). Genes shown in the figure are indicated in bold. Notably, in the Meth-SIVE derived cells, expression of the microglia activation marker gene *AIF1* (aka *Iba1*) was elevated 2.4-fold, whereas the M2 macrophage marker gene *CD163*, which is expressed on microglia in SIVE and HIVE [61,62], as well as perivascular macrophages recruited in the late stages of SIVE [63], was decreased 2.9-fold in expression. The M2 macrophage marker *STAB1* was also reduced in Meth-SIVE 3.0-fold relative to SIVE. In addition, the homeostatic microglia marker *P2RY12* was also decreased 6.6-fold. The pan-macrophage marker gene *CD14* was decreased in expression in Meth-SIVE 1.6-fold, whereas another pan-macrophage marker, *CD68*, was increased in Meth-SIVE 1.5-fold. In Meth-SIVE, the tissue-resident macrophage marker gene factor XIII A chain (*F13A1*) was elevated in expression by 1.8-fold, whereas another tissue-resident macrophage marker gene, growth arrest-specific 6 (*GAS6*, 3.4-fold) was lower. Expression of the colony stimulating factor 1 receptor gene *CSF1R*, aka macrophage colony-stimulating factor receptor, crucial for the maintenance of microglia in the brain [64], was decreased 2.0-fold in Meth-SIVE. The relative expression of these marker genes indicates that the microglia/macrophages in Meth-SIVE animals differed from those in SIVE animals, with the main changes being a decrease in markers of M2 macrophages and an alteration in the pattern of activation markers.

#### 3.3.2. Pathway Analysis

We next examined the entirety of the genes that were differentially expressed between the Meth-SIVE condition and SIVE (Appendix A). To assess the impact of these changes, we used Ingenuity Pathway Analysis (IPA) to map the changes in gene expression onto known patterns in diseases/biological functions, as well as canonical pathways (Appendix A). Among the top diseases/biological functions that were increased, Meth-SIVE relative to SIVE was organismal survival, with organismal death, morbidity, and mortality having high Z-scores (6.8 and 6.7), respectively. Cell death and survival were also notable, with apoptosis and necrosis having Z-scores of 2.3 and 1.8, respectively. Regarding pathways, Meth-SIVE macrophage/microglial cells were elevated in the PPARα/RXRα Activation pathway (Z-score 2.1), Pentose Phosphate pathway (PPP, Z-score 1.6), Complement system (Z-score 1.6), and p53 signaling (Z-score 1.5).

#### 3.3.3. Differential Gene Expression

Examination of genes whose expression differed between Meth-SIVE and SIVE, as well as their expression within the graph-based clusters, revealed many important clues as to the changes resulting from meth in the presence of CNS disease (Figure 6, genes shown in the figure are indicated in bold in the text). Many complement components are known to be made by microglia, and the complement pathway is known to participate in synaptic pruning. Consistent with the pathway analysis above, expression of complement ***C3*** (4.4-fold), as well as *C1QB* (2.9-fold) and *C1QC* (4.0-fold) genes was decreased in Meth-SIVE. Interestingly, gene expression of complement factor D (***CFD***, aka adipsin), part of the alternate complement activation pathway, was increased in Meth-SIVE (7.3-fold). The S100 proteins comprise a family of calcium-binding proteins, and the gene expression of many were upregulated in Meth-SIVE, including ***S100A4*** (18.3-fold), *S100A2* (30.2-fold), *S100A6* (14.5-fold), *S100A8* (3.6-fold), and *S100A9* (4.3-fold). While their protein products can have a wide variety of functions, macrophages recruited to the CNS in SIVE are marked by expression of S100A9 [65]. While expression of the TIMP metallopeptidase inhibitor 1 (***TIMP1***) gene was greatly increased in Meth-SIVE (21.2-fold), none of the other members of the TIMP family, nor any members of the matrix metalloproteinase family (MMPs), were found to be differentially regulated. **TIMP1** has been proposed as a neuroprotective factor in the HIV-infected brain [66]. The osteopontin (***SPP1***) gene was expressed at high levels in SIVE but down-regulated in Meth-SIVE (3.2-fold). **SPP1** is an inflammatory cytokine that is elevated in the brains of those with HIVE, as well as in the cerebrospinal fluid of HIV-infected individuals irrespective of CNS disease [67]. Its production is thought to help limit neuroinflammation during HIV infection [68].

We next examined genes encoding proteins that affect cellular signaling and its regulation, as well as transcription factors (Figure 7, genes shown in the figure are indicated in bold in the text.) Triggering receptor expressed on myeloid cells (TREM) are membrane proteins that help signal proinflammatory responses. The expression of the ***TREM2*** gene, whose product is thought to be protective in Alzheimer’s disease through promotion of amyloid clearance [69], was down-regulated in SIVE-Meth 3.3-fold, whereas *TREM1* gene expression was increased 2.1-fold. TREM1 and **TREM2** often have opposing effects, thus the reciprocal patterns are consistent with this finding, as TREM1 can amplify immune responses, whereas **TREM2** can suppress them [70]. Regulators of G-protein signaling (RGS) proteins attenuate signaling by G-protein α-subunits. The genes of a number of members of this family were expressed in SIVE but down-regulated in Meth-SIVE: ***RGS1*** (4.6-fold), *RGS2* (2.4-fold), *RGS10* (3.3-fold), and *RGS16* (6.8-fold). All have been shown to have a role in macrophage functions. Another negative regulator of signaling is dual-specificity protein phosphatase 1 (**DUSP1**), which inactivates key signaling intermediates, mitogen-activated protein kinases. ***DUSP1*** gene expression is fairly highly expressed in both conditions, but is 2.2-fold lower in Meth-SIVE. Expression of two other DUSP genes was lower in Meth-SIVE (*DUSP10*, 2.5-fold and *DUSP16*, 1.9-fold), whereas expression of two was higher (*DUSP2*, 3.5-fold and *DUSP5*, 1.8-fold). Transcription factors are the most important regulators of gene transcription, and many showed differential expression between SIVE and Meth-SIVE cells. These included expression of the *RXRA* gene, increased 3.3-fold in Meth-SIVE, consistent with the findings of increases in the PPARα/RXRα activation pathway found above. **SPI1** (aka PU.1,) is a master transcriptional regulator critical for macrophage development [71], and ***SPI1*** gene expression is upregulated 1.9-fold in Meth-SIVE. Gene expression of the transcription factor *SALL1* is down-regulated 7.1-fold in Meth-SIVE. *SALL1* has been characterized as a microglial signature gene, and its loss of expression results in the conversion of microglia to an inflammatory phenotype [72]. A number of members of the Krüppel-like factor (KLF) family of zinc-finger transcription factors are differentially expressed. Gene expression of ***KLF2*** was down-regulated in Meth-SIVE (2.2-fold), as was *KLF6* (2.2-fold), whereas *KLF13* was increased 2.8-fold.

#### 3.3.4. Upstream Regulators

To further characterize the observed expression changes in the DEGs, we identified the potential upstream regulators that could yield such effects. IPA upstream regulator analysis was used here for this purpose. It identifies potential regulators based on curated prior knowledge of expected effects between transcriptional regulators and the targets. More specifically, IPA upstream regulator analysis predicts a relevant transcriptional regulator through examining how many targets are present in our data and compares their direction of changes to what is expected from the literature; if the direction of changes is mostly consistent with a particular activation state, a prediction is made with two statistical measures, the *p*-value of overlap and activation Z-score [73].

Using IPA, a number of regulators were identified (Appendix A). The brain-derived neurotrophic factor (BDNF) was identified as an important upstream regulator, and its mechanistic pathway was predicted to be inhibited (negative activation Z-score of 3.5) (Figure 8). In addition, CREB-binding protein was predicted as an upstream regulator, also with a negative activation Z-score (2.7). PPARα was predicted to be a positive upstream regulator, with an activation Z-score of 2.6, consistent with the pathway analysis above.

While its actions are prominent in the CNS, BDNF can be detected in the blood, and its level has been examined in a number of neurological and neuropsychiatric conditions [74,75,76,77]. Prominent among these is depressive disorder, in which BNDF is lower than in control subjects [78]. To examine the relationship of BDNF levels to meth treatment, we assessed BDNF concentrations in a separate cohort of rhesus monkeys that had been treated with meth (*n* = 7), or as a control, saline (*n* = 6). After eight weeks of maintenance meth treatment, plasma BDNF was significantly lower in meth-treated animals than in controls (Figure 9). While the source of BDNF in the plasma is not clear (from the CNS or peripheral sources), BDNF can be transported bidirectionally across the blood–brain barrier [79], and blood BNDF levels have been found to reflect brain levels [80].

### 3.4. Analysis of Microglial Clusters

As evident in Figure 10A (and see Figure 3A,B), Cluster 1 was predominately cells from SIVE, clusters 3 and 4 predominately cells from the Meth-SIVE, and clusters 2 and 5 were shared between SIVE and Meth-SIVE. Hierarchical clustering (Figure 10B) revealed that clusters 1 and 2 have patterns quite analogous to each other; similarly, clusters 3 and 4 were alike. Cluster 5 combined features from both, largely lacking the heightened expression of differentially expressed genes, which can also be seen in the gene expression of *RGS1*, *DUSP1*, and *KLF2* (see Figure 7).

To gain additional insight into these clusters of cells, we performed IPA to determine unique upregulated pathways enriched by the differentially expressed genes (Appendix A). Clusters 1 and 4 contained many such pathways, and examples include, in Cluster 1, the Neuroinflammation Signaling Pathway (Z-score 4.0, Figure 11) and Unfolded Protein Response (Z-score 3.2); and in Cluster 4, Glycolysis (Z-score 2.2), as well as the PPP (Z-score 2.4). Fewer pathways were enriched in Clusters 2 and 3. Cluster 5 also had fewer enriched pathways, but was notable for enrichment in genes involved in Oxidative Phosphorylation (Z-score 5.2, Figure 12), the most efficient means to generate ATP for energetic use.

## 4. Discussion

Using single microglia and macrophages from the brains of monkeys with productive SIV infection, we have found that meth significantly increases the proportion of infected cells within the CNS. A number of genes were differentially expressed in these cells between the meth-treated monkeys with SIVE and those with SIVE that were not treated with meth. These were notable for decreases in markers of M2 macrophages, alterations in the expression patterns of several activation markers, and changes in the expression of a number of key regulatory genes. Mapping these changes in gene expression onto functions revealed indicators of death and disease. Bioinformatic analysis also revealed upstream targets that may be modulated to reverse such changes. Examples of this modulation included activation of the PPARα pathway and inhibition of CREB-binding protein and the BDNF pathway.

BDNF has been characterized for its role in many functions, including neurogenesis, neuronal survival, synaptogenesis, and synaptic transmission [81,82]. However, while many of these functions are ascribed to its production by neurons, microglial-derived BDNF has been shown to support learning-related synapse formation, and thus promote learning and memory [83]. BDNF levels have been found to be inversely related to microglial activation, as evidenced in vivo in aging and both in vivo and in vitro by BDNF treatment, reducing microglial activation; the production of HIV-associated neurocognitive disorders persist in the era of potent antiretroviral therapy, and CHARTER Study inflammatory mediators [84]. In a mouse microglial cell line, increases in the transcription factor KLF2, which we found to be decreased in microglia in meth-treated animals, increased microglial BDNF expression [85], further supporting the role of the upstream BDNF pathway in regulating microglial function. Interestingly, in rats, acute meth treatment increased BDNF expression in the brain, whereas chronic meth treatment lowered BDNF in the brain [86,87]. The BDNF pathway thus represents a potential target in alleviating the untoward effects of meth on the brain in the context of HIV infection. Interestingly, BDNF is thought to be reduced in HAND/neuroHIV, and enhancement of BDNF levels has been proposed as a treatment for this condition [88].

Understanding highly dimensional data, such as scRNA-seq, is aided by both visualization algorithms such as UMAP and clustering algorithms such as graph-based methods. In the context of encephalitis, microglia/macrophages from the meth-treated animals indeed separated from those without meth treatment, however, the SIV-infected cells did not differ appreciably from the uninfected ones by these techniques. This is similar to our recently reported study of in vitro HIV infection of mature monocytes, in the presence or absence of antiretroviral treatment, where uninfected cells did not cluster separately from the infected ones, but gene expression was significantly altered in the infected cells compared to the uninfected cells [55].

While we did find several clusters that helped distinguish cells from Meth-SIVE and SIVE, we note that all of the cells were exposed to the inflammatory environment, thus the ability to identify differences with resting microglia/macrophages was not possible, although given the expression of many activation makers, such distinctions are likely. Two of the clusters were of particular interest. Cluster 4, comprised largely of cells from Meth-SIVE, was enriched for glycolysis and PPP. These are parallel pathways; glycolysis begins with the conversion of glucose to glucose-6-phosphate and results in ATP production, whereas PPP starts with glucose-6-phosphate and leads to the production of the reducing agent NADPH, as well as precursors for other biochemical pathways. In myeloid cells, such as macrophage and microglia, NADPH oxidase then produces reactive oxygen species (ROS) to combat pathogens. The PPP was also increased in the overall comparison of Meth-SIVE to SIVE. Cluster 5, containing cells from both Meth-SIVE and SIVE, also had enrichment for a metabolic pathway. These cells were highly enriched in expression for genes encoding components of the oxidative phosphorylation pathway. Alterations in metabolism in immune cells, known as immunometabolism, has received increasing investigation in correlations with function, including in macrophage–pathogen interactions and inflammation [89,90,91,92,93]. Recent data indicate that shifting ATP production from oxidative phosphorylation to the glycolysis pathway is required to activate the inflammasome, leading to production of both interleukin 1 beta (IL-1β) and ROS [93,94,95,96]. Notably, expression of the inflammasome-associated genes *IL1B*, *NLRP3*, and *PYCARD* was increased in cells from the Meth-SIVE animals relative to cells from the SIVE animals (Appendix A). This suggests that this is a pathway by which meth could enhance neuroinflammation, and represents an intriguing area of future study regarding neuroHIV.

There is increased microglial activation in the brains of people who abuse meth [97]. Cellular and tissue context is important, as treatment of rodents with meth increases the amount of inflammatory cytokines in the brain, but no changes in cytokine levels were seen in isolated microglia treated with meth in vitro [5]. This suggests that the effect of meth on microglia is indirect, requiring a factor present in the intact brain. The inflammatory effects of meth are likely tied to dopamine release, which occurs in neurons exposed to meth, but not in purified cultured microglia. This is supported by studies in which experimental reduction of striatal dopamine blocks meth-induced microglial activation [98], as well as the finding that meth-induced microglial activation was found in the striatum, where dopaminergic nerve terminals release dopamine, but not in substantia nigra, where the dopaminergic cell bodies reside [99]. The role of dopamine is further supported by rodent studies showing that pharmacologically blocking or knocking out D1-like dopamine receptors prevents methamphetamine associated increases in proinflammatory cytokines, microglial activation, and neurotoxicity in the striatum [4,100,101]. In addition to the direct effects of dopamine, dopamine is also crucial in a second well-described mechanism, that of oxidative stress. The heightened brain dopamine release resulting from meth administration leads to increased oxidation and production of ROS (in addition to that resulting from the PPP through NADPH oxidation, as described above), resulting in cellular damage and glial activation [102,103].

We previously reported a study in which rhesus monkeys were infected with SIV, and after reaching stable, steady-state viremia, were divided into a meth treatment and control group at 19 weeks post-infection [40]. Then, they received a similar meth ramp-up (over five weeks) and maintenance dosing (23 weeks) to the animals in the current study, but for a longer duration. No SIV-specific neuropathology was found, however, significantly more virus was found in the brains of the meth-treated animals than in the brains of the control animals. Here we found a significant difference in the number of SIV-infected cells, which were increased in the meth-treated animals. While we did not find a difference here in viral load in brain regions between the meth-treated and non-meth treated animals, in the current study, the animals had SIVE with viral loads in brain regions ranging from ~10^6^–10^8^ RNA genome equivalents/µg tissue RNA, whereas in the prior study the viral loads in brain regions ranged from ~10^1^–10^3^ RNA genome equivalents/µg tissue RNA. The lack of a difference due to meth in the current study could be due to a ceiling effect given the high levels of virus, a result of the small numbers of animals examined, a difference in the meth administration time (here before infection, in the prior study after control of viral load and for an extended time), or a true lack of effect in the setting of encephalitis.

As mentioned above, one limitation to our study is sample size. The concern over small sample size is shared with many scRNA-seq experiments, since, as in other studies, this can decrease statistical power, yielding a type II error. Here we made use of archived samples, and to collect more prospectively would require the use of many more monkeys, as only a proportion (one-fourth to one-third) typically develop SIVE. We note that in addition, type I errors can of course still exist. Thus, as in all science, our findings can be examined through replication and other means of validation in future studies. Such subsequent work can also include determining the effects of meth in uninfected animals and in infected animals with viral loads controlled with antiretroviral therapy, as well as findings in humans.

## 5. Conclusions

Meth treatment of rhesus monkeys led to distinct changes in microglia and brain macrophages in animals that developed SIVE. ScRNA-seq demonstrated that a higher proportion of these myeloid cells were infected in the presence of meth. Overall, cells from the Meth-SIVE animals overexpressed genes in pathways involved in morbidity and cell death, and deficiencies in the BDNF-signaling pathway were predicted. SIV-infected cells did not cluster, based on gene expression patters, differently in animals that were treated or not with meth. However, there were distinct differences, as well as similarities, in the clustering of brain macrophages and microglia due to meth. One such cluster predominately contained cells from animals without meth treatment, and was enriched in expression of genes in the neuroinflammation signaling pathway, whereas another cluster predominately contained cells from animals receiving meth treatment and was enriched in expression of genes involved in glycolysis as well as the PPP. In addition, a separate cluster contained cells from animals in both conditions and was enriched for expression of genes in the oxidative phosphorylation pathways. Thus, meth is associated with increased infection of CNS myeloid cells and alters their metabolic pathways. The BDNF pathway was identified to be inhibited by meth, and represents a potential target for treatment. By examining on a single-cell level the response of a large number of cellular targets of HIV/SIV and meth in a gold-standard model, this study illustrates a unique productive approach to a more complete understanding of the interactions of drugs of abuse and neuroHIV, in addition to identifying potential strategies to alleviate these deleterious effects.

## Figures and Tables

**Figure 1 viruses-12-01297-f001:**
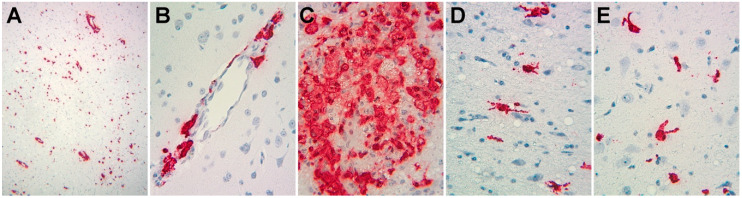
Photomicrographs of in situ hybridization study of SIV RNA expression in the brains of animals from this study. Signals representing SIV RNA are red, nuclei counterstained with hematoxylin are blue. (**A**) Low power (original magnification 100×) from animal 17T, revealing abundant expression. (**B**–**E**) Medium power (original magnification 200×) revealing cells with the morphological appearance of perivascular macrophages (**B**, from animal 21T), macrophage clusters (**C**, from animal 32T), and cells with the morphological appearance of activated microglia (**D**,**E**, from animal 34T).

**Figure 2 viruses-12-01297-f002:**
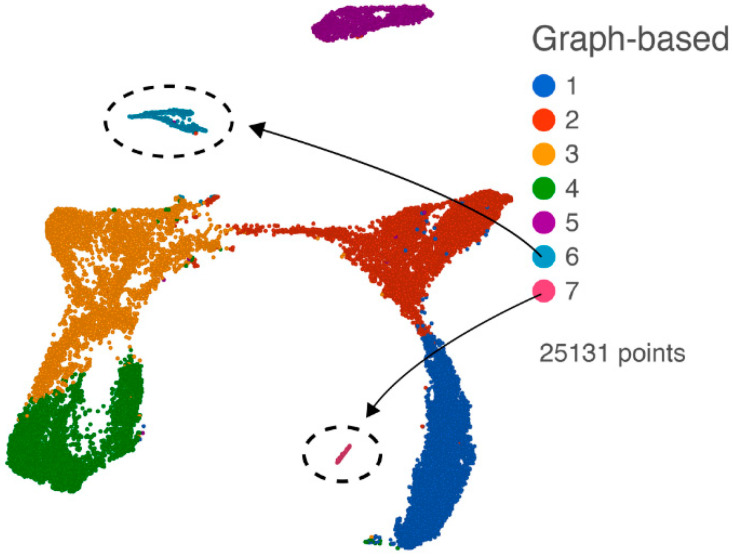
UMAP plot of all cells following QA/QC, normalization, and filtering with the annotated gene list. Graph-based clusters are designated by different colors, and cluster 6 (CTL and NK cells) and cluster 7 (endothelial cells) are indicated.

**Figure 3 viruses-12-01297-f003:**
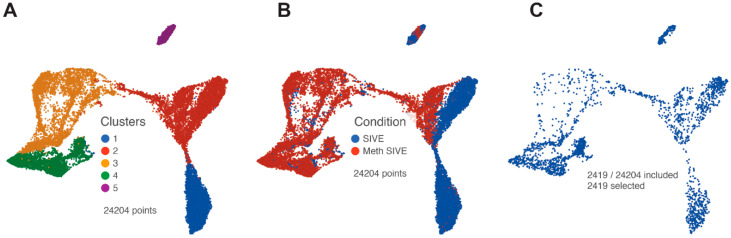
UMAP plots after removal of clusters containing CTL/NK and endothelial cells. (**A**) Graph-based clusters are designated by different colors. (**B**) Cells from the two conditions are designated by different colors. (**C**) Only cells that contained RNA transcripts from SIV are shown.

**Figure 4 viruses-12-01297-f004:**
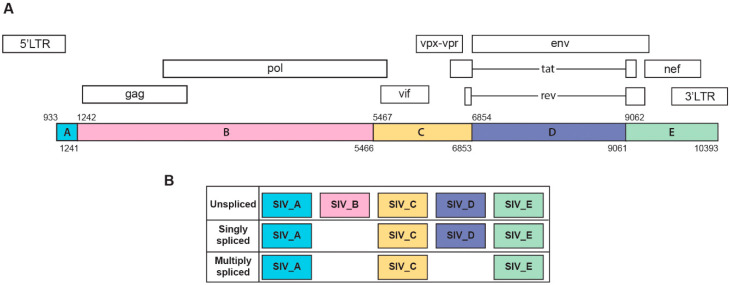
(**A**) Schematic representation of SIV genome and its annotations. (**B**) Characterization of SIV transcripts based on major splicing patterns. This figure was created based on a modification of those found in [55,60].

**Figure 5 viruses-12-01297-f005:**
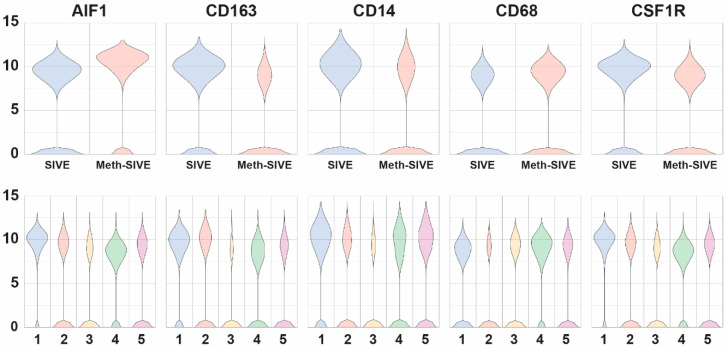
Violin plots showing the relative expression of macrophage/microglia marker genes in the two different conditions (upper) and the five different clusters (lower).

**Figure 6 viruses-12-01297-f006:**
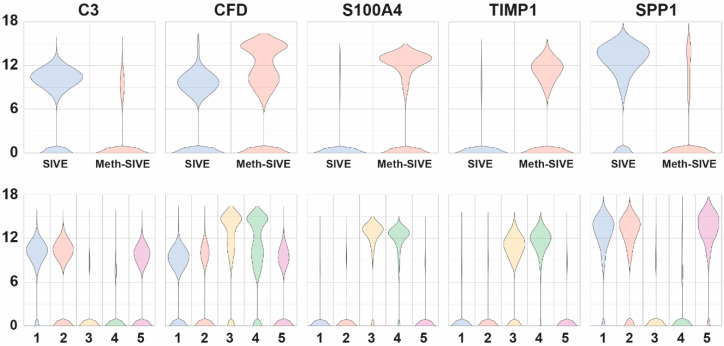
Violin plots showing relative expression of genes that are differentially expressed between SIVE and Meth-SIVE in the two different conditions (upper) and the five different clusters (lower). The genes shown are bolded in the text.

**Figure 7 viruses-12-01297-f007:**
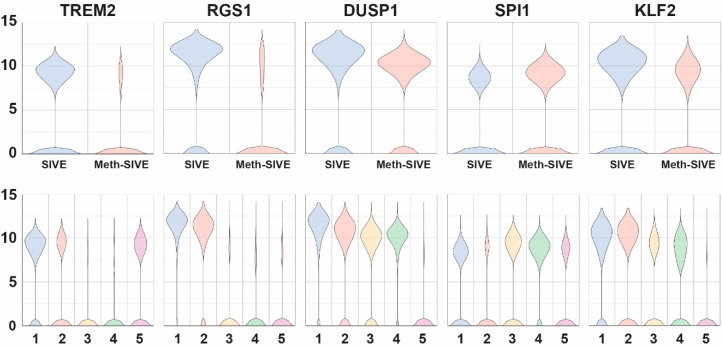
Violin plots showing the relative expression of regulatory genes in the two different conditions (upper) and the five different clusters (lower). The genes shown are bolded in the text.

**Figure 8 viruses-12-01297-f008:**
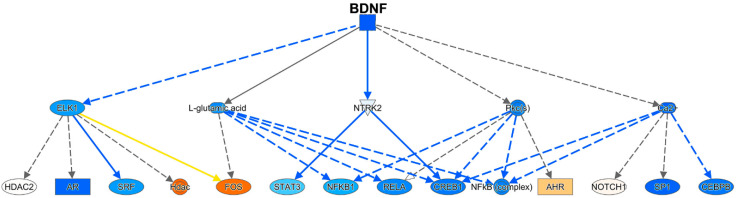
BDNF pathway. The colors of the symbols correspond to levels of expression or activity, with darker shades of orange indicating an increase in Meth-SIVE, and darker shades of blue decreasing in Meth-SIVE. Solid lines signify direct interactions, and dashed lines signify indirect interactions. Blue lines denote inhibitory interactions, yellow denote findings inconsistent with the state of the downstream molecule, and black lines denote that the nature of the effect is not predicted. Figure adapted from an IPA-generated diagram.

**Figure 9 viruses-12-01297-f009:**
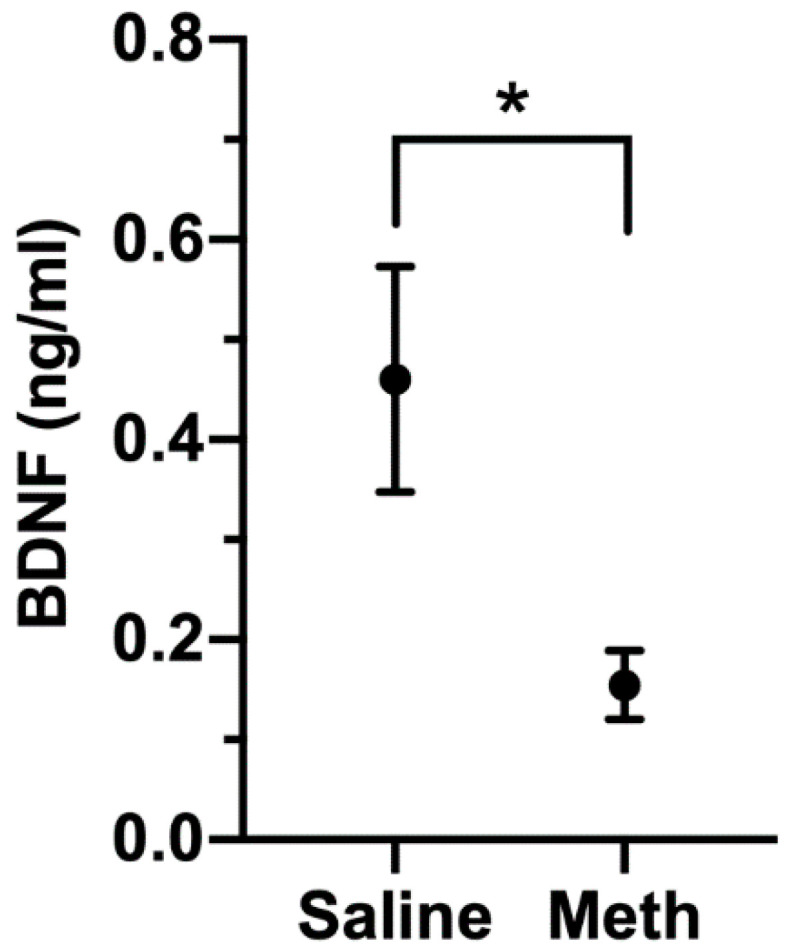
Plasma BDNF levels in saline (*n* = 6) and meth (*n* = 7) treated monkeys. One of the meth-treated animals had levels below the limit of quantification, and the value of the limit of quantitation (0.045 ng/mL) was used for analysis. Figure shows mean and SEM, * *p* = 0.039, Mann–Whitney U test.

**Figure 10 viruses-12-01297-f010:**
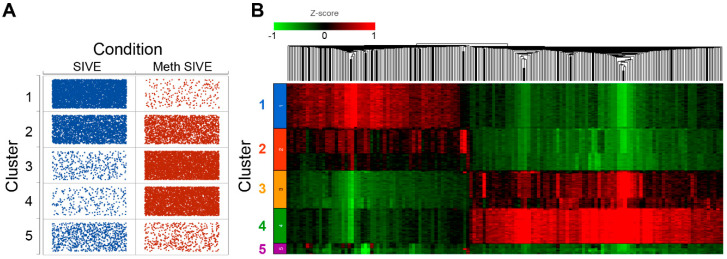
(**A**) Dot plot showing the number of cells (dots) in each cluster from each condition. (**B**) Heat map of hierarchical clustering of 1753 DEGs between SIVE and Meth-SIVE within the clusters, colored by expression level Z-score.

**Figure 11 viruses-12-01297-f011:**
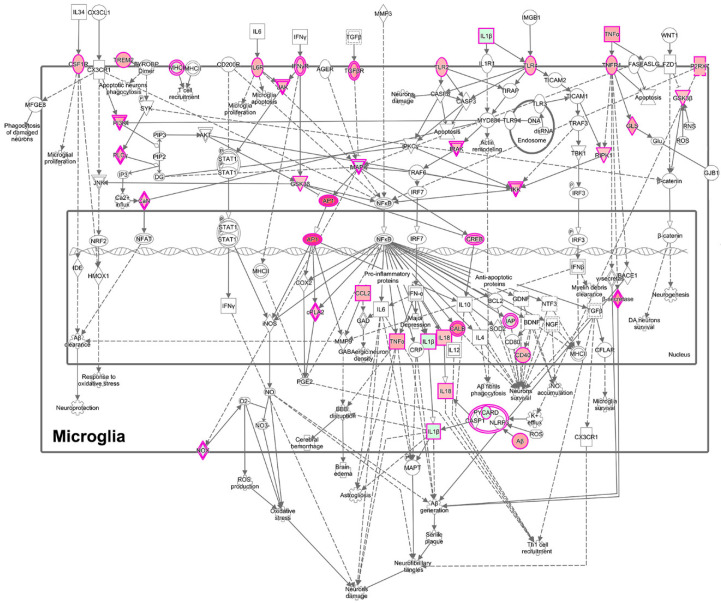
Enrichment of the neuroinflammatory pathway in microglia in cluster 1, based on the DEGs between cluster 1 and clusters 2–5. Cluster 1 is comprised predominately of cells from SIVE animals. The pink border indicates molecules with expression data, and the double border indicates a group or complex of molecules. Orange fill indicates increased expression, green decreased expression. Figure adapted from an IPA-generated diagram.

**Figure 12 viruses-12-01297-f012:**
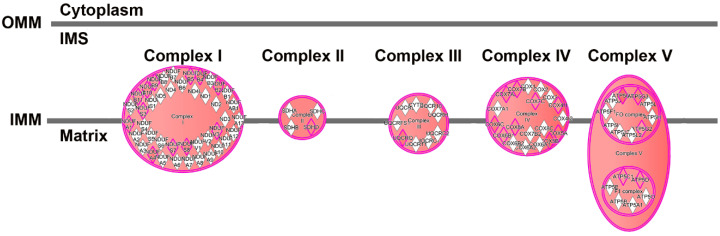
Enrichment of genes involved in the electron transport chain in cluster 5, based on the DEGs between cluster 5 and clusters 1–4. Pink border indicates molecules with expression data, double border indicates a group or complex of molecules. Orange fill indicates increased expression of the molecules and of the double border complexes. Figure adapted from an IPA-generated diagram.

**Table 1 viruses-12-01297-t001:** Specifics on the animals used in this study.

Animal	Age (Yrs)	Condition	Days p.i.	Terminal Plasma Viral Load (log_10_/mL)	Tissue Viral Load (log_10_/µg RNA)			
					Caudate	Frontal Cortex	Hippo-Campus	Spleen
17T	5.1	SIVE	65	8.8	6.3	7.2	7.2	8.2
34T	2.7	SIVE	92	7.8	7.8	8.1	7.7	7.8
21T	5.4	Meth-SIVE	92	8.5	7.5	7.3	6.6	7.5
32T	4.0	Meth-SIVE	78	7.7	7.3	7.7	7.6	8.0

Animals used for isolation of brain macrophages/microglia, including age at necropsy and condition (meth indicates pre-infection ramp-up and maintenance for 60 days followed by inoculation with SIV, and then maintained on meth until necropsy). Days p.i. is time from SIV inoculation to necropsy. Viral loads were determined as described in Methods.

**Table 2 viruses-12-01297-t002:** Distribution of SIV transcripts by region in cells classified by condition or cluster.

Region	Bases	SIVE	Meth-SIVE	Cluster 1	Cluster 2	Cluster 3	Cluster 4	Cluster 5
SIV_A	309	3.34%	4.22%	2.16%	7.01%	1.85%	3.80%	5.18%
SIV_B	4225	4.20%	5.85%	2.92%	6.95%	3.15%	7.44%	5.93%
SIV_C	1387	1.59%	1.97%	0.96%	4.07%	0.77%	1.19%	1.98%
SIV_D	2208	1.81%	2.23%	1.12%	4.17%	0.86%	1.61%	3.07%
SIV_E	1332	6.21%	7.58%	3.70%	9.13%	4.61%	10.39%	9.00%
SIV_All		8.48%	11.33%	5.62%	11.58%	7.31%	16.36%	10.91%

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
