# Peer review of "Methamphetamine Increases the Proportion of SIV-Infected Microglia/Macrophages, Alters Metabolic Pathways, and Elevates Cell Death Pathways: A Single-Cell Analysis"

_viruses, 2020, doi:10.3390/v12111297_

Round 1

Reviewer 1 Report

This paper describes the single cell analysis of brain-derived macrophages and microglia in methamphetamine, SIV infected macaques. The results offer a powerful view of cellular responses to these stimuli. But the results are somewhat limited by the small number of animals used, and the failure to show evidence of animal-to-animal variation. While this is clearly explainable by the expense of macaques, it is clearly possible that with groups of two, differences between groups could arise from a single animal that differs from the other three. This would be especially true of the main finding, that meth causes a higher number of cells to be infected.

Two minor editorial comments:

  1. Starting line 188: Different typeface and spacing
  2. In Fig 10 A the cluster numbering has cluster one at the bottom, while in 10B it is at the top. Please change one so that that they have the same orientation and align each cluster to its mate in the other panel. These changes will make it easier for the viewer to grasp differences and similarities in the clusters.

Author Response

Thank you for the thoughtful review. We appreciate the comment on the sample size. Therefore we have added the paragraph below to the end of the Discussion section. As far as the numbered points:

  1. The different font issue has been resolved.
  2. We agree, that change in Figure 10 facilitates our conveying information to the reader. Since the dot plot (panel A) is of fixed height, whereas the hierarchical clustering (panel B) is proportional height, we could not align exactly, but have indeed reversed the order in A so the it is the same as B, and adjusted he vertical positioning to align as best as possible.

The added paragraph at the end of the Discussion:

     One limitation to our study is sample size. This may affect the possibilities of false negatives, such as the above viral load issue. The concern over small sample size is shared with many scRNAseq experiments, as with other studies it can decrease statistical power yielding such a type II error. Here we made use of archived samples, and to collect more prospectively would require the use of many more monkeys, as only a proportion (one-fourth to one-third) typically develop SIVE. We note that type I errors can of course still exist. Thus, as in all science, our findings can be examined through replication and other means of validation in future studies. Such subsequent work can also include determining the effects of Meth in uninfected animals, in infected animals with viral loads controlled with antiretroviral therapy, as well as findings in humans.

Reviewer 2 Report

In this manuscript, Niu et al undertake a thorough study of the single cell transcriptome of cells of the macrophage/microglial lineage in the brains of four SIV-infected rhesus macaques that were exhibiting SIV-associated encephalitis. Two of these had been treated chronically with methamphetamine both before and after SIV inoculation. They confirm that a higher proportion of microglia/macrophages were infected in the meth-treated animals, and that meth treatment had resulted in significant shifts in transcriptional profile. These changes indicated both a shift in relation to canonical M1/M2 polarization and in relation to the microglia-expressed BDNF pathway, arguably of relevance in neuronal health and memory acquisition.

While interesting, these results are, in my view, less compelling than they might have been, had it been possible to include four parallel samples from non-SIV-infected animals, which would have enabled a more mechanistically interesting analysis of the direct effects of meth treatment, SIV-induced neuroinflamamtion, and the interaction of the two.

Author Response

We thank the reviewer for their thoughtful review. We agree that additional experiments would add to our understanding of the effects of methamphetamine and SIV on microglia and neuroinflammation in the primate system. To address this, and the comments by Reviewer 1, we have added the following paragraph at the end of the Discussion:

            One limitation to our study is sample size. This may affect the possibilities of false negatives, such as the above viral load issue. The concern over small sample size is shared with many scRNAseq experiments, as with other studies it can decrease statistical power yielding a such a type II error. Here we made use of archived samples, and to collect more prospectively would require the use of many more monkeys, as only a proportion (one-fourth to one-third) typically develop SIVE. We note that type I errors can of course still exist. Thus, as in all science, our findings can be examined through replication and other means of validation in future studies. Such subsequent work can also include determining the effects of Meth in uninfected animals, in infected animals with viral loads controlled with antiretroviral therapy, as well as findings in humans.